# Development of UAV Tracing and Coordinate Detection Method Using a Dual-Axis Rotary Platform for an Anti-UAV System

**Bor-Horng Sheu [1], Chih-Cheng Chiu [2], Wei-Ting Lu [1], Chu-I Huang [1] and Wen-Ping Chen [1,3,*]**

[1] Department of Electrical Engineering, National Kaohsiung University of Science and Technology, Kaohsiung City 80778, Taiwan

[2] Engineering Science and Technology, College of Engineering, National Kaohsiung University of Science and Technology, Kaohsiung City 80778, Taiwan

[3] Ph.D Program in Biomedical Engineering, Kaohsiung Medical University, Kaohsiung City 80778, Taiwan

* Correspondence: wpc@nkust.edu.tw

**Abstract:** The rapid development of unmanned aerial vehicles (UAVs) has led to many security problems. In order to prevent UAVs from invading restricted areas or famous buildings, an anti-UAV defense system (AUDS) has been developed and become a research topic of interest. Topics under research in relation to this include electromagnetic interference guns for UAVs, high-energy laser guns, US military net warheads, and AUDSs with net guns. However, these AUDSs use either manual aiming or expensive radar to trace drones. This research proposes a dual-axis mechanism with UAVs automatic tracing. The tracing platform uses visual image processing technology to trace and lock the dynamic displacement of a drone. When a target UAV is locked, the system uses a nine-axis attitude meter and laser rangers to measure its flight altitude and calculates its longitude and latitude coordinates through sphere coordinates to provide drone monitoring for further defense or attack missions. Tracing tests of UAV flights in the air were carried out using a DJI MAVIC UAV at a height of 30 m to 100 m. It was set up for drone image capture and visual identification for tracing under various weather conditions by a thermal imaging camera and a full-color camera, respectively. When there was no cloud during the daytime, the images acquired by the thermal imaging camera and full-color camera provide a high-quality image identification result. However, under dark weather, black clouds will emit radiant energy and seriously affect the capture of images by a thermal imaging camera. When there is no cloud at night, the thermal imaging camera performs well in drone image capture. When the drone is traced and locked, the system can effectively obtain the flight altitude and longitude and latitude coordinate values.

**Keywords:** unmanned aerial vehicles (UAVs)**,** dynamic coordinate tracing; computer vision; anti-UAV defense system

## 1. Introduction

Since Dà-Jiāng Innovations (DJI) released the new DJI XP3.1 (a flight control system) [1,2], its multi-rotor drone has been characterized by hovering, route-planning, its comparative low price, and ease of use [3,4]. These factors have led to the rapid development of drones in the consumer selfie drone market. At present, they have been adopted all over the world in environmental aerial photography or surveillance. However, under uncontrolled drone flight control, there have been many drone misfortunes, both at home and abroad. News events have included the impact on the navigation area, the impact on famous buildings, and the horror of political figures, and the biggest worry for drones is that people or terrorists may modify them to make them carry dangerous bombs,

or use the vehicles themselves as extremely dangerous devices for attacking bombs. Drone potential hazards are an urgent problem [5,6], so countries have begun to impose strict regulations on drone traffic management, and even implement no-fly rules. The Federal Aviation Administration (FAA), similar authorities of China and Taiwan have all drawn up drone system usage scenarios. Besides, for the sake of preventing and controlling indecent invasions by drones, research into many AUDSs has progressively turned into an international research topic of interest [7–9]. There are lots of similar approaches of AUDS, such as spoofing attacks, tracing and early warning, detection, defeating, and signal interfering [10,11]. The scalable effects net warhead invented by the US military launches warheads and throws nets to attack enemy drones, and other anti-UAV technologies developed by scientists. Among them, devices such as radar, satellites, and thermal imaging cameras are used for early warning and detection, which detect the trace of a drone from the ground. Those technologies are to carry out an immediate detection, warning, and tracing of the intruding drone, and provide real-time messages and information support [12–15]. Destroying intruding drones is another kind of approach, which acquires the location of drones first and then defeats them by using missiles, lasers, or other weapons. An example of this type of approach is a defense system using strong energy laser developed by a Turkish company ASELSAN. Although the attack method has the effect of destruction and intimidation, there is a risk of accidental injury to nearby people if the UAV is shot down. Interfering with communications of an intruding drone by electromagnetic interference methods is effective; however, it also affects the surrounding communications. Thus, the shortcoming is clear—the wireless communications around that area is affected. The quality of life will also be influenced. The scalable effects net warhead is a technique for quickly and accurately delivering a net to a remote target using a projectile. It includes a mesh-containing launcher, an ejection spring, and a releasable ogive. This net warhead can be launched from a suitable barrel using a propellant or compressed gas. The speed and rotation of the warhead will be determined by the diameter of the barrel. When the warhead approaches the target, the electronic control board activates the servo motor and then pulls the central locking plunger to release the ogive. Then, the spring pushes the net to aim at and spread on the target resulting in its capture.

There are lots of reports of security issues caused by drone incidents. Lots of incidents reported that terrorists around the world use drones to attack the targets. Thus, there are an increasing market and insistent need for AUDS, which lead to the development of AUDS technologies. However, with limited resources, how to develop AUDS detection technologies for civil purposes is a very important research topic. This research proposes a dual-axis rotary tracing mechanism, which adopts a thermal imaging camera and a full-color camera. With an algorithm of image identification, this device traces and locks a drone. The device traces a drone while the drone is in the range of visual detection by tracing of dynamic coordinates. A nine-axis attitude meter and laser rangers in the proposed device are used to perform the sensing then calculate its flight altitude of the drone. The spherical coordinates are used to obtain the longitude and latitude data of the drone. Through continuous locking the dynamic coordinates of the drone, a ground control station arranges an attack drone to defeat the target drone by a net gun, laser gunm or other weapons such as rifles. In this stage, a real-time transmitting of the dynamic coordinate to an attack drone for fast chase is required. This paper proposes a different approach.

## 2. System Architecture

The research structure of this paper consists of three units: The dual-axis mechanism, drone identification, tracing approach, and drone three-dimensional coordinate calculation method. A detailed description of each unit follows.

### 2.1. The Dual-Axis Rotary Platform Mechanism

The design of externals for the proposed dual-axis rotary device in this research is shown in Figure 1. There are two axes: The lateral axis is the *x*-axis (yaw) and the longitudinal axis is the *y*-axis

(pitch). With the rotation of yaw and pitch, the movement of the image (the picture acquired by the thermal imaging or full-color camera) will be driven. The image provides flying drone detection with a viewing angle range of 360° for yaw rotation and 180° for pitch rotation. In order to improve the detection capability of the drone, there is a mechanism frame inside the dual-axis mechanism, which is equipped with a fine-tuning device (three sets of laser rangers), a nine-axis attitude meter, a thermal imaging camera, and a full-color camera for drone image capture and position detection. The fine-tuning mechanism is used to provide drone tracing, and the laser ranger is adopted for focusing precision adjustment. In the drone image focus and distance measurement, because the image acquisition lens and the laser rangers are not on the same line, the distance measuring of the drone for high-altitude flight needs the fine-tuning mechanism for the focus. First, the photographic lens is adopted as the focus point of the center of the image frame, then the three laser rangers are set close to the lens, before the fine-tuning mechanism is modified with the focus distance of the drone to be acquired by the tracing device, so that the hitting spot of the laser ranger on the drone overlaps with the spot of the center of the image. GPS and LoRa antennas are placed outside the tracing device due to the need to avoid metal disturbance and prevent from signal-shelter in the thick casing of the tracing device.

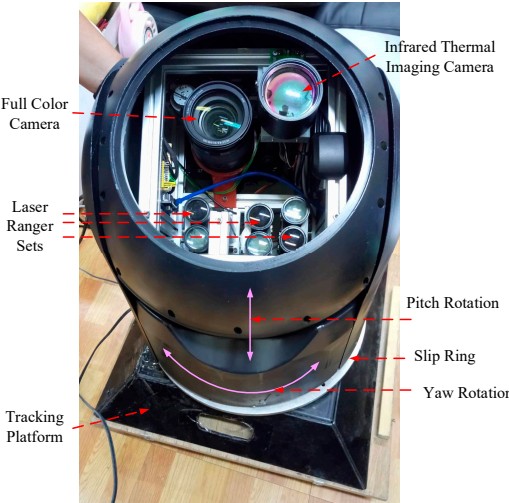

**Figure 1.** Design of the dual-axis mechanism.

## 2.2. Drone Visual Image Tracing Method

The drone image visual tracing process identifies a drone in motion using images acquired by a thermal imaging camera or full-color camera. These cameras inside a dual-axis mechanism were adopted for image acquisition in this study. After image acquired, vision image processing is performed and then drone tracing is done through motor control of the above platform.

A dual-axis tracing device adopts image processing techniques to deal with images of a dynamic object. An image capture card in the device acquires pictures at a speed of 30 frames per second, and then performs image processing, as shown in the image processing flowchart in Figure 2. The acquisition of a dynamic drone contour [15,16] is carried out by the frame difference method with image pre-processing.

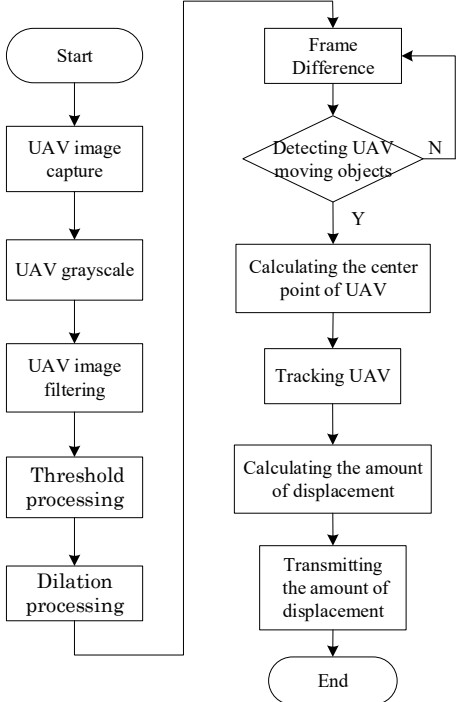

**Figure 2.** Drone image tracing flowchart.

First, the original image, as shown in Figure 3a, is converted to grayscale of $S_{gray}$ by RGB conversion [15] as shown in Figure 3b. The formula is:

$$S_{gray} = 0.299R + 0.587G + 0.114B \tag{1}$$

where RGB is the three primary color pixel values of the original image. The grayscale image of Gaussian filter and median filter is then used to remove noise and smooth the image to avoid misjudging subsequent images due to noise effect as shown in Figure 3c. Processing Figure 3c as a histogram can obtain a histogram of the image. The histogram of the image can be found that there is a highest value in the histogram. Its value is the T value. After threshold processing, a binary image can be obtained as shown in Figure 3d.

After threshold processing is performed, the method of dilation in morphology can be used to segment independent image element of the UAV so that the UAV is highlighted in the picture for better tracking of the system. The dilation operation is shown in Equation (2), where A is the original image and B is the mask. The core operation is that there is an object at the neighboring point and the position is set as an object.

$$A \oplus B = \{z | (\hat{B})_z \cap A \neq 0\} \tag{2}$$

Figure 3e shows the image after dilation (morphology), which fills the gap in the broken UAV image. The frame difference method allows extraction of a motion region [17,18] on two adjacent images by performing differential operations. It oversees the movement of anomalous objects [19] in the picture with the movement of the target and the movement of the camera. There will be a significant difference between adjacent frames. For real-time tracing, due to the fast drone flight speed, a small amount of immediate calculation is needed to trace the drone steadily. There is an advantage that the background accumulation problem for the frame difference method is almost ignored. The renew speed is quick, and less calculation is required for tracing a drone immediately, which is shown in Figure 3f. The equation is as follows:

$$D(x,y) = f_{m+1}(x,y) - f_m(x,y) \tag{3}$$

where $D(x, y)$ is the result of differential operation for the $m^{\text{th}}$ frame and $(m+1)^{\text{th}}$ frame.

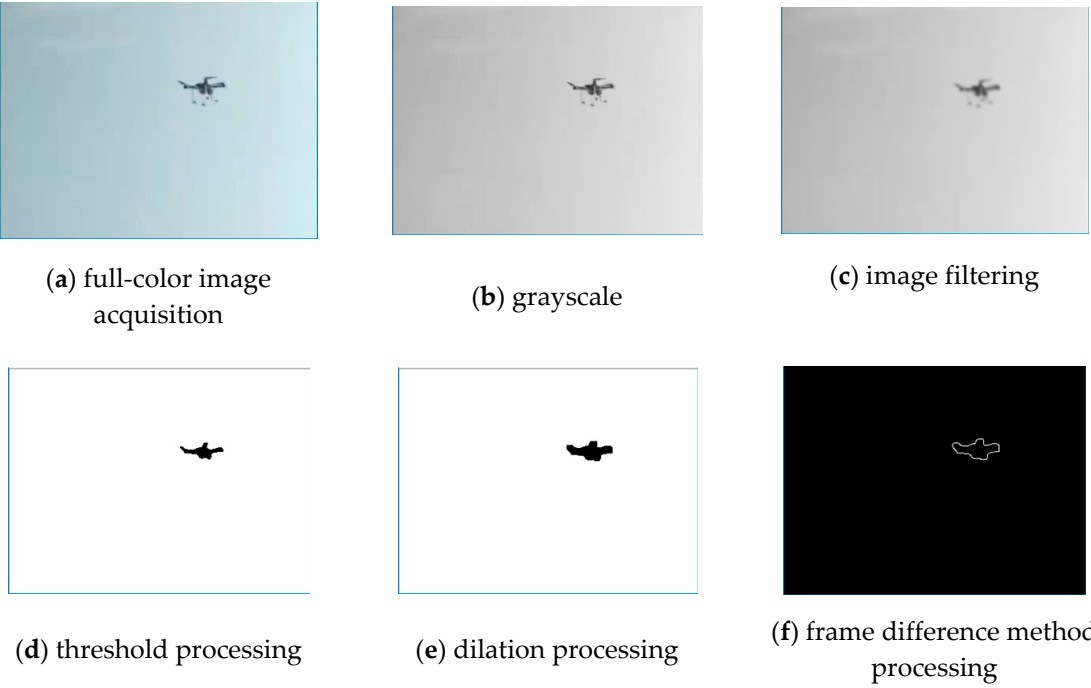

**(a)** full-color image acquisition

**(b)** grayscale

**(c)** image filtering

**(d)** threshold processing

**(e)** dilation processing

**(f)** frame difference method processing

**Figure 3.** Drone image acquisition procedures.

The drone tracing mechanism of a dual-axis tracing device is to convert a two-dimensional variation of the pixels of a captured image ($\Delta$x, $\Delta$y) into a step-level control amount of a two-axis step motor inside the above device. A movement amount is calculated by the above conversion of acquired images, which is applied to complete drone tracing operations. Suppose the image's resolution is set as 2P × 2Q pixels, which is shown in Figure 4. An acquired screen image's origin point is set as $(x_0, y_0)$, and its pixel coordinate is (-P, -Q). The screen's center point is $(x_1, y_1)$, which is the concentric point in the tracing screen, and its pixel coordinate is defined as (0, 0). Therefore, the pixel of the screen image in the $x$-axis ranges from -P~+P, and the $y$-axis ranges from −Q~+Q. When the target drone's image enters the edge of the rectangular frame of the tracing device, it is acquired by the visual identification of the system. The tracing system instantly estimates the pixel difference $(p, q)$ between the $x$-axis and the $y$-axis of the center position $(x_2, y_2)$ and the concentric point $(x_1, y_1)$ of the drone: $(x_2 − x_1, y_2 − y_1)$. When the p value is positive, it reveals that the center point of the target drone is on the right side of the center point of the screen image, and vice versa for the left side. Furthermore, when the q value is positive, it reveals that the center point of the target drone is on the lower side of the center point of the screen image, and vice versa.

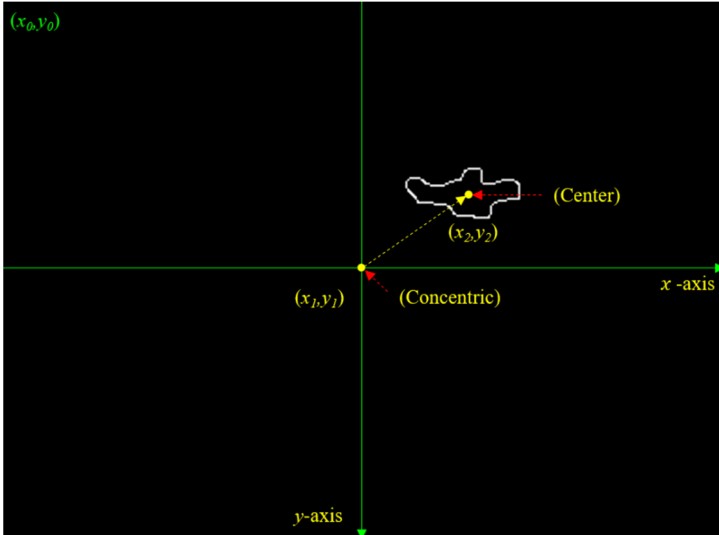

**Figure 4.** Diagram of the target drone displacement vector.

In order to trace the invading drones dynamic position at any time, three-dimensional coordinate parameters such as the flight altitude of the drone and its longitude and latitude are calculated, and these parameters are immediately transmitted to the ground monitoring center for direct attack or indirect hunt for the drone. The control requirements for a two-axis step motor in the dual-axis tracing device are as follows:

Suppose the position of the concentric point of the screen image is $(x_1, y_1)$ and the position of the drone is $(x_2, y_2)$, which are shown in Figure 4. The system traces the drone in real time, and the requirement of motor control speed of the dual-axis device is:

$$V_{yaw-rotaion} = \frac{p}{\Delta t} \ and \ V_{pitch-rotaion} = \frac{q}{\Delta t} \tag{4}$$

where $\Delta$t is the time for the duration of the image capture card to acquire images, which is normally 30 frames per second; that is, $\Delta t = 1/30$ s. The smaller the value of $\Delta$t, the more accurately the tracing system can grasp the drone movement trajectory, so that the alignment screen's concentric point of the tracing device can lock the drone as soon as possible, but relatively requires a faster software computing requirement.

The system has to convert the pixel difference $(p, q)$ of the image into the tracing step angle of the two-axis step motor of the tracing device; that is, the image coordinates are converted from pixel values $(p, q)$ to the number of steps $(S_p, S_q)$ to drive the two-axis step motor, where $Sp$ and $Sq$ represent the step numbers of the step motor for the yaw rotation and pitch rotation, respectively. Since the step motor rotates 360° in one turn, assuming that $S_R$ is the total number of steps in one circle of the step motor, the step angle $\theta_m$ of each step motor is:

$$\theta m = \frac{360^0}{S_R}. \tag{5}$$

Due to the fact that the necessary $\theta_m$ value of the step motor in the tracing device is related to the pixel value acquired by the image capture card, it is important to find the correlation between these two matters. First, suppose the full-color (or thermal imaging) camera acquires the picture in a certain direction as shown in Figure 5a. Then, suppose the driver of the step motor in the tracing device send the square wave signal with the number of $S_P$ to the step motor of the *x*-axis, which drives the step motor of the *x*-axis to carry out the rotation of S$_P$ steps. These make the dual-axis device carry out a horizontal right movement, and then acquires the immediate picture, as shown in Figure 5b. Finally, by comparing the images of Figure 5a,b with the superimposed image and calculating the

increment of the p-value of these two images of the *x*-axis after the step motor of the *x*-axis rotates steps of $S_p$, as shown in Figure 5c, the tracing system can receive the number of steps $p_S$ of the step motor necessary for each pixel in the yaw rotation direction:

$$p_S = \frac{S_p}{p}. \tag{6}$$

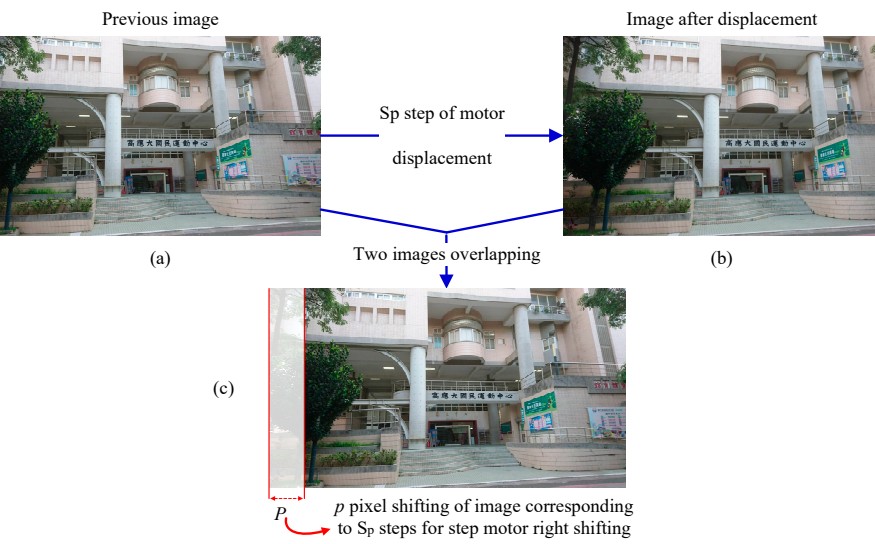

**Figure 5.** Schematic diagram of the relationship between the step angle and image.

Furthermore, the number of steps of the step motor necessary for each pixel in the pitch rotation direction is $q_S$.

$$q_S = \frac{S_q}{q} \tag{7}$$

*2.3. Drone Three-Dimensional Coordinate Calculation*

(1) Detection method for drone height: The rotation direction and angle judgment mode of the dual-axis tracing device are based on the value sensed from nine-axis attitude meters. Therefore, there is a close relationship between the placement position of the attitude meter and the accuracy of the system data. The starting point of orientation determination by an attitude sensing component is based on the preset position of its own module. The drone movement orientation of the screen image is based on the movement of the lens, so the position of the sensing component of the attitude meter should be placed as close as possible to the camera lens, so that the angular deviation will be smaller. After many tests, we found that it has the best effect when placed under the lens.

When the tracing device finds that the drone has entered the visual image of the screen as in Figure 6, the system immediately traces the drone image in the screen through visual identification, and computes the relative relationship between the drone image and the center point of the visual image screen. Then, it performs an action of target drone tracing and position locking. The drone image enters the center point $(x_1, y_1)$ of the visual image screen of the tracing device, and the laser ranger can measure when the target drone enters the laser ranging range. The distance value ($r$) from the target is obtained, and the pitch angle (θ) of the rotating stage at this time is obtained by the nine-axis attitude meter in the apparatus. As shown in Equation (7), the flight altitude h of the drone can be converted.

$$h = r \times sin\theta = \sqrt{(x_2 - x_1)^2 + (y_2 - y_1)^2} \times sin\theta \tag{8}$$

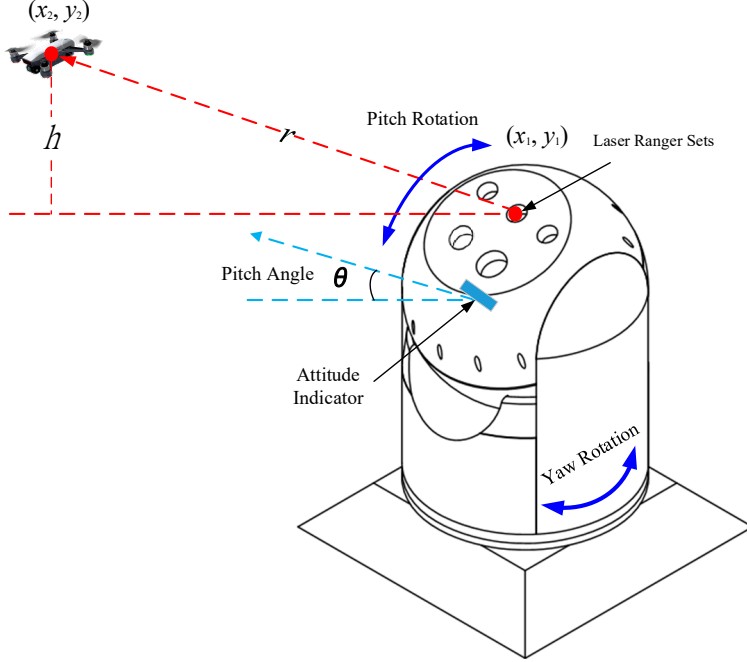

**Figure 6.** Schematic diagram of the calculation of the height of the drone on the image concentric point by the attitude meter and the laser ranger.

(2) Detection method for drone longitude and latitude coordinate as well as azimuth: When the tracing system locks the target drone to the center point position $(x_1, y_1)$ of the image screen through tracing, the target drone has entered the laser ranging range. The values of the distance and the horizontal elevation angle of the target drone from the test site can be obtained by laser rangers and the attitude meter of the dual-axis device, and then the longitude and latitude, and the azimuth angle (relative to the north angle), of the drone can be analyzed. In this paper, the drone longitude and latitude coordinates [20,21] can be obtained by using the target coordinate conversion equation. A schematic diagram of the longitude and latitude coordinate algorithm of the target drone is shown in Figure 7. This algorithm is applicable to the distance calculation between any two points on the spherical surface, where O is the center point of the sphere, and A and B are any two points on the sphere. Point A represents the longitude and latitude coordinates $(W_A, J_A)$ obtained by the GPS in the dual-axis tracing platform, and point B represents the unknown longitude and latitude coordinates $(W_B, J_B)$ of the drone traced by the dual-axis tracing platform. The longitude and latitude coordinates of point B can be obtained by the sphere longitude and latitude coordinate conversion [21]. Other parameters are as follows:

Point G: The true north position of the sphere; point O: Center of the sphere; R: Average radius of the earth (~6371 km);

$(W_A, J_A)$: The values of the longitude and latitude of point A obtained by the GPS of the dual-axis device;

ψ (Azimuth angles of point A, B and G): Obtained by the magnetometer and gyroscope of the nine-axis attitude meter in the dual-axis device;

∠a: The angle between the two points B and G and the ground connection $\overline{GO}$;

∠b: The angle between the two points A and G and the ground connection $\overline{GO}$;

∠c: The angle between the two points A and B and the ground connection $\overline{GO}$;

$\hat{AB}$: The spherical distance between the two points of point A (dual-axis device) and B (drone aerial location); that is, the length of minor arc $\hat{AB}$ in the arc generated by the intersection of the planes passing through the three points A, O, and B, and the ball.

∠c is the metric angle:

$$\angle c = \frac{\hat{AB}}{R} \times \frac{180^0}{\pi}. \tag{9}$$

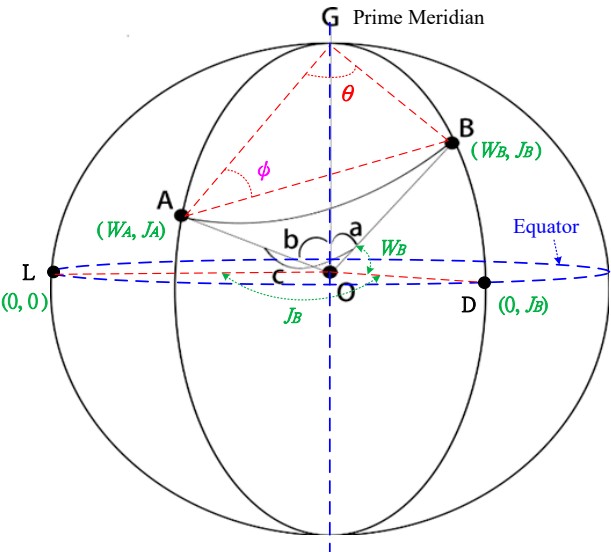

**Figure 7.** Schematic diagram of the longitude and latitude sphere coordinate algorithm for the drone.

Since the spherical distance between point A (dual-axis device) and point B (drone aerial location) is much smaller than the length of R, the spherical distance $\hat{AB}$ between two points can be approximated as the linear distance between two points $\overline{AB}$, so Equation (9) can be corrected to:

$$\angle c = \frac{\overline{AB}}{R} \times \frac{180^0}{\pi}. \tag{10}$$

∠*a* and ∠θ can be converted by the sphere longitude and latitude coordinates [21], and Equation (11) and Equation (12) can be obtained as follows,

$$\angle a = cos^{-1}(sinA_w \times cos\angle c + cosA_w \times sin\angle c \times cos\psi), \tag{11}$$

$$\angle\theta = sin^{-1}\left(\frac{sinc \times sin\psi}{sina}\right). \tag{12}$$

Since the drone flight altitude at point B is within a few kilometers above the ground, the length between the two points A and B and the point O of the center of earth can be regarded as the same, and ∠θ can be regarded as the longitude difference between the two points A and B. Thus, the value of the longitude and latitude of point B ($W_B, J_B$) is the longitude and latitude coordinates of drone, where:

$$J_B = J_A + \angle\theta. \tag{13}$$

As described above, the longitude and latitude values of the drone flying in the air, which is locked by the dual-axis device and the flight altitude h, can be calculated and acquired, then transmitted wirelessly to the ground control center for the attack tasks of direct attack or tracing and entrapping.

## 3. Experimental Results and Discussion

The drone tracing test results of the dual-axis device, software, and hardware used in the study are as follows:

*3.1. Development Environment of the Software and Hardware*

(1) Visual identification software development: This study uses OpenCV (open-source computer vision library), which is a library application software for computer vision and machine learning. It was originally initiated and developed by the Intel Corporation and distributed under the license of BSD (Berkeley Software Distribution license). It can be used free of charge in business and research. OpenCV includes many image processing functions, machine learning algorithms, and libraries for computer vision applications.

(2) Sensors and imaging equipment:

- Infrared thermal imaging: As shown in Figure 8, this paper uses an intermediate level infrared thermal imaging camera (TE–EQ1), which is equipped with a 50 mm fixed-focus lens with 384×288 pixels. It can be used for mobile tracing tests with a drone flying altitude of about 100 m. The TE–EQ1 is capable of observing the distribution of heat sources in drones, providing users with drone tracing tasks through computer vision technology.

- Full-color single-lens reflex camera: This study uses a Sony $\alpha$7SII (Japan) with a Sony SEL24240 lens, as shown in Figure 9. The Sony $\alpha$7SII features ultra-high sensitivity and an ultra-wide dynamic range. It has a 35 mm full-frame 12.2 megapixel image quality, with a wide dynamic range of ISO 50 to 409,600 and a BIONZ X processor, which optimizes the performance of the sensor, highlights details, and reduces noise, and records 4K (QFHD: 3840*2160) movies in full-frame read-out without image pixel merging, effectively suppressing image edge aliasing and moiré. The main reason for choosing this device and lens is that the Sony $\alpha$7SII can be transmitted via HDMI, achieving high-quality image instant transmission and less delay in camera transmission than others of a similar price, allowing instant acquisition of images and image processing. Additionally, the Sony SEL24240 telephoto single lens has a focal length of 240 mm, allowing drones at short distances (<100 m) to be clearly displayed on the tracing screen.

- Thermal image acquisition: The UPG311 UVC image acquisition device is used to read the infrared thermal imaging images. The infrared thermal imaging output interface is NTSC or PAL. It is compatible with multiple systems and supports plug-and-play and UVC (USB video device class). The agreement supports an image resolution of up to $640 \times 480$ pixels.

- Frame grabber for full-color camera: We used the Magewell USB Capture HDMI Gen2 capture card, which supports single-machine simultaneous connection of multiple groups of operations, and is compatible with Windows, Linux, and OS X operating systems with no need to install drivers or a real Plug and Play. It supports a completely standard development interface. In addition, the input and output interfaces use HDMI and USB 3.0, respectively, it supports an input image resolution of up to $2048 \times 2160$ and a frame rate of up to 120 fps, and it automatically selects the aspect ratio that is the most appropriate for the picture.

- Laser ranger: This article uses the LRF 28-2000 semiconductor laser ranger, as shown in Figure 10. The main reason for using this laser ranger is that it uses an RF section of 900~908 nm wavelength to protect the human eye, with a range of 3.5 m to as long as 2.0 km. Its measurement resolution and measurement accuracy are 0.1 m and 1.0 m, respectively. Its specifications are suitable for the measurement of the flight altitude of the short-range drones in this study, and its significant distance measurement accuracy can be used as a basis for testing the drone flight altitude.

- Multirotor: At present, the market share of multirotors is dominated by DJI Dajiang Innovation. With its drone, which is lightweight, portable, and has a small volume being sold at a friendly price, it is very popular among people generally. Additionally, the news media have reported that most drone events have involved drones from DJI Dajiang Innovation, so this study uses the DJI Mavic Pro multirotor as the main tracing target of the experiment.

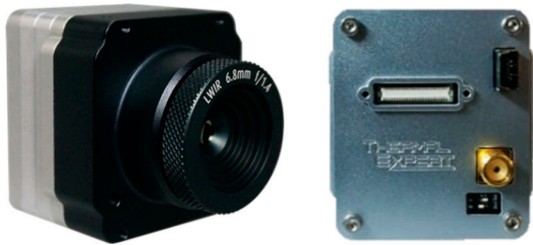

**Figure 8.** TE–EQ1 infrared thermal imaging device.

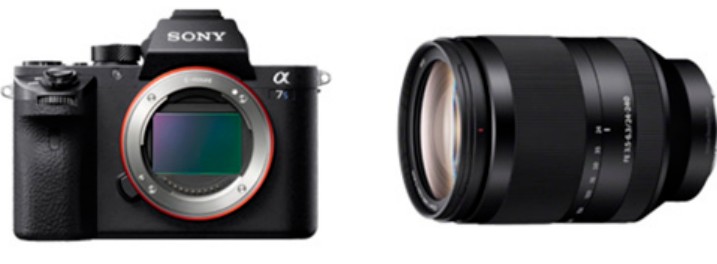

(**a**) Body: Sony $\alpha$7SII       (**b**) Lens: Sony SEL24240

**Figure 9.** Single-lens reflex camera.

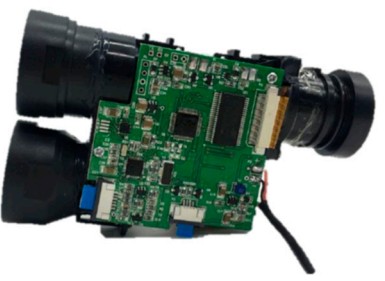

**Figure 10.** Laser Ranger LRF 28-2000B.

### 3.2. Development of Hardware for the Dual-Axis Device

To make it shockproof, robust, and waterproof, the outer material of the dual-axis device of this paper is made of Teflon. It is equipped with two sets of step motors (including its step motor driver). These two-step motors respectively control the up and down elevation angle and the movement of the left and right rotation of the dual-axis device. The pixels of full-color image used in this device are 640 × 480; that is, P = 640 and Q = 480, and a two-phase step motor $S_R = 10,000$ is used; that is, $\theta_m = 0.036$. The test height of drone tracing is 10~100 m, and the radius of the dual-axis device is a = 0.25 m. The maximum flight speed of the drone is 20 km/h. As a result, the maximum rotation speed of the dual-axis device is $V_{yaw,max} = \frac{a}{R_{min}} \times V_{uav} \approx 0.139 \, (m/s)$.

Figure 11a shows the dual-axis tracing device and Figure 11b shows the hardware structure inside the dual-axis tracing device. Figure 11c shows the focus fine-tuning structure diagram for the laser ranger.

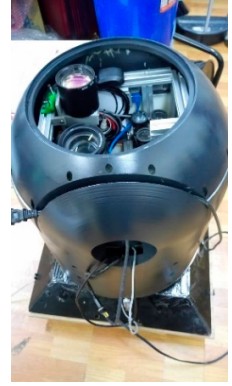 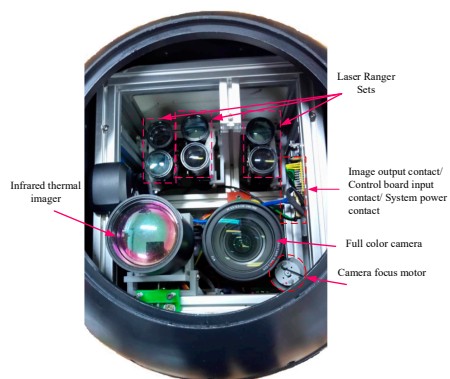

(**a**) The dual-axis tracing device    (**b**) Appearance of the front view of the image lens/laser ranger

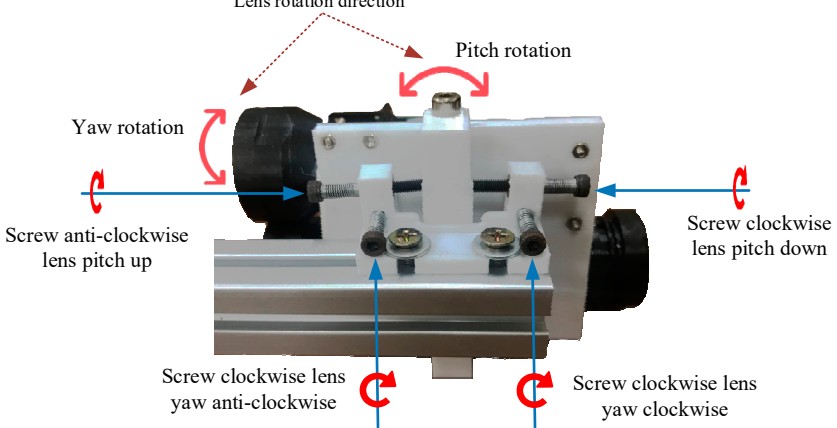

(**c**) Picture of the focus fine-tuning configuration of the laser ranger

**Figure 11.** The inner hardware structure of the dual-axis tracing device.

The main control board of the dual-axis tracing device is shown in Figure 12a and the motor control unit is shown in Figure 12b.

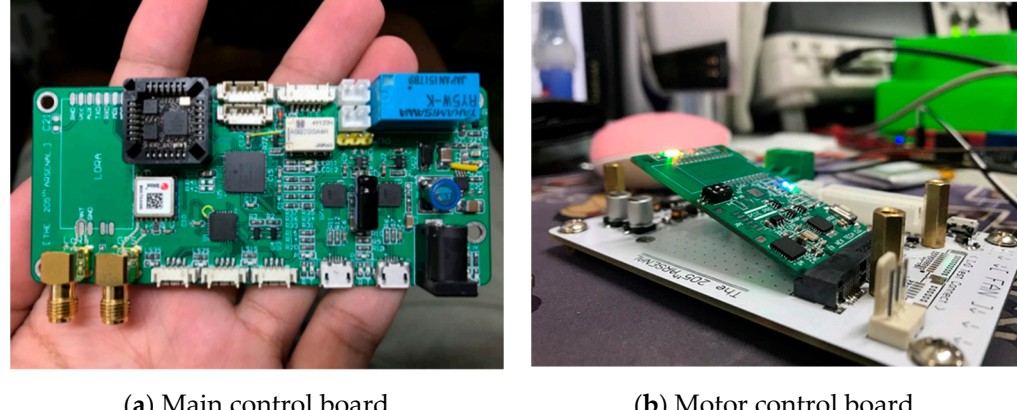

(**a**) Main control board        (**b**) Motor control board

**Figure 12.** 3D simulation and physical circuit picture of the main control unit and motor control unit.

### 3.3. Drone Image Identification and Tracing Test in Different Weather Environments

This paper uses the DJI Mavic Pro four-axis drone to achieve tracing tests of drone flight in all kinds of weather. The results of tracing tests on sunny days and at flight altitudes of about 70–105 m

are shown in Figure 13. At more than 102 m, drone movement cannot be determined (restricted by the resolution of the full-color camera). The test results of various weather conditions—-sunny, cloudy, and rainy—- are shown in Figures 14–16. The detailed test description is as follows:

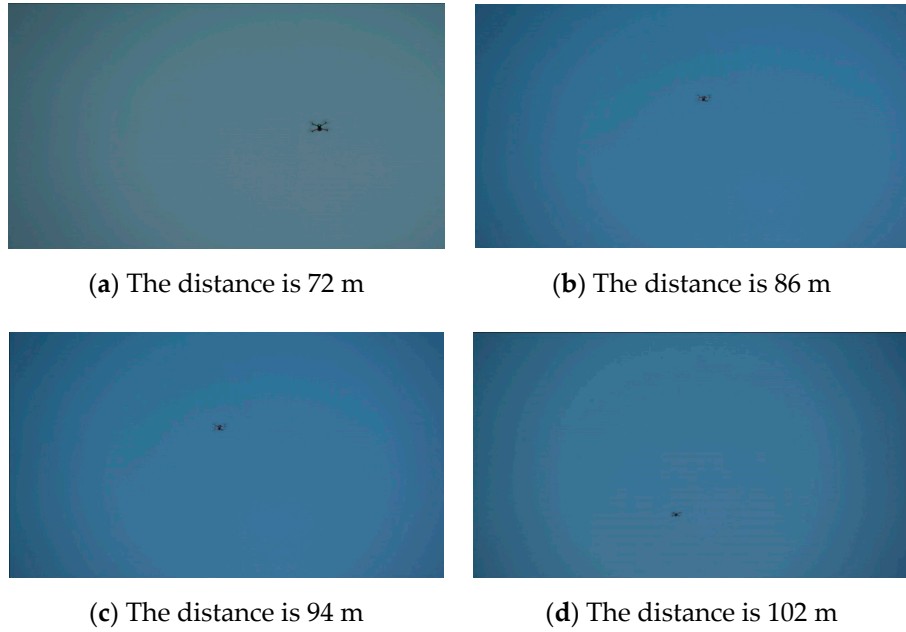

(**a**) The distance is 72 m        (**b**) The distance is 86 m

(**c**) The distance is 94 m        (**d**) The distance is 102 m

**Figure 13.** Tests of drone flight altitudes (on a sunny day).

(1) Sunny environment: As shown in Figure 14, the acquisition effects of the drone's image on sunny days are different based on different brightness and sunshade levels due to different conditions. In the case of large changes in ambient brightness, the effect of light on the lens and tracing is quite large. In addition, the size of the drone is small, which can cause the phenomenon that the target drone is lost while in image processing, and then recovered.

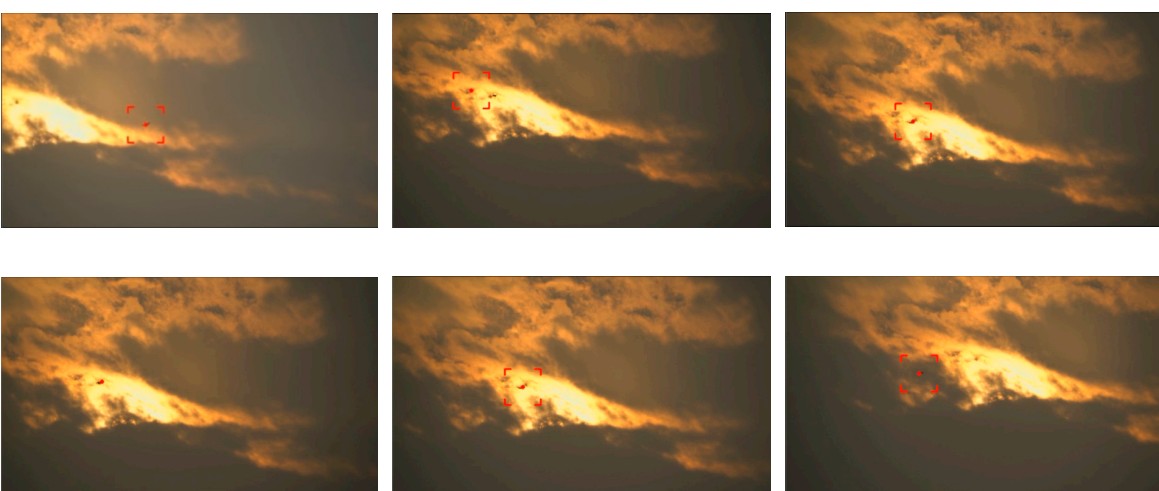

**Figure 14.** Results of the tracing test on sunny days (drone flight altitude is 105 m).

(2) Cloudy environment: Figure 15 shows the acquisition effects of the drone's image on cloudy days. The cloudy background is clean and free of other interference. In addition, the difference in grayscale presented by the sky is not obvious, making the tracing target more significant. The tracing effect was good, and the target drone was not easy to lose.

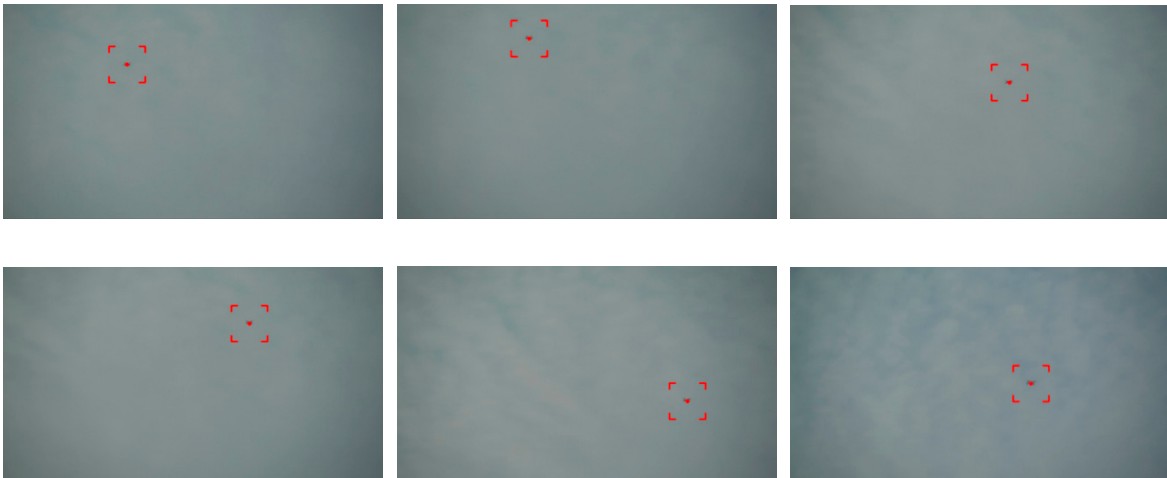

**Figure 15.** Results of the tracing test on cloudy days (drone flight altitude is 130 m).

(3) Rainy environment: Figure 16 shows the acquisition effects of the drone's image on rainy days. In general, no drone appears on rainy days (a normal drone is usually not waterproof); drone players usually choose to take off in clear weather. In the case of rain, the lens is susceptible to raindrops. Coupled with poor vision, it is impossible to clearly present the tracing image, and it is difficult to judge the target, which leads to a poor tracing effect.

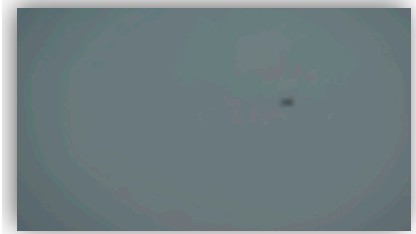

**Figure 16.** Results of the tracing test on rainy days (drone flight altitude is 80 m).

*3.4. Test of Drone Tracing and Longitude and Latitude Coordinates*

Figure 17 shows the tracing process through a full-color image. A target drone was successfully hit with laser rangers in this study. Conversion of the sensed values of the nine-axis attitude meter and the known longitude and latitude coordinates ($W_A$, $J_A$) as (120.328766, 22.648851) of the GPS of the dual-axis device were achieved through the longitude and latitude sphere coordinate algorithm. It can be calculated that when the drone is locked by the tracing system, its current longitude and latitude coordinates ($W_B$, $J_B$) (based on Equations (12) and (13)) are (120.328766, 22.648394), and the flying altitude is 58.7 m.

Figure 18 shows the tracing process through a thermal image at the same place. A target drone was successfully hit with the laser rangers of the dual-axis device. Conversion of the sensed values of the nine-axis attitude meter and the known longitude and latitude coordinates ($W_A$, $J_A$), where a longitude and latitude coordinate (120.328766, 22.648851) of the GPS of the device were achieved through the longitude and latitude sphere coordinate algorithm. It can be calculated that when the drone is locked by the tracing system, its current longitude and latitude coordinates ($W_B$, $J_B$) (based on Equations (12) and (13)) are (120.328697, 22.648524) and the flying altitude is 51.9 m.

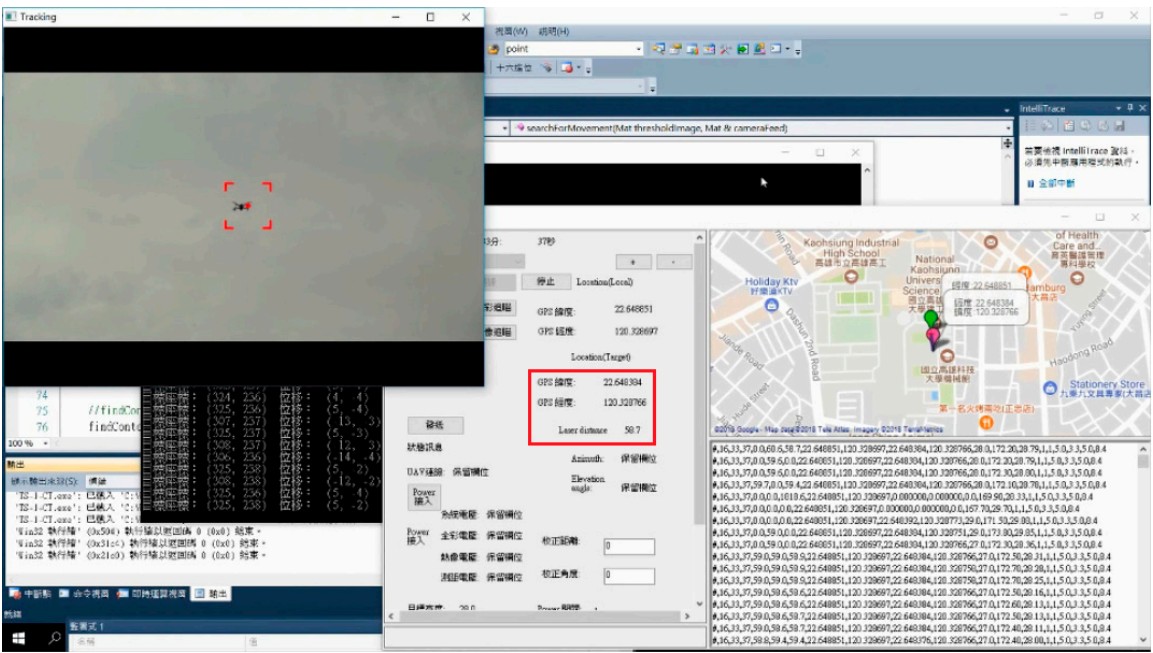

**Figure 17.** Test results of the full-color tracing interface.

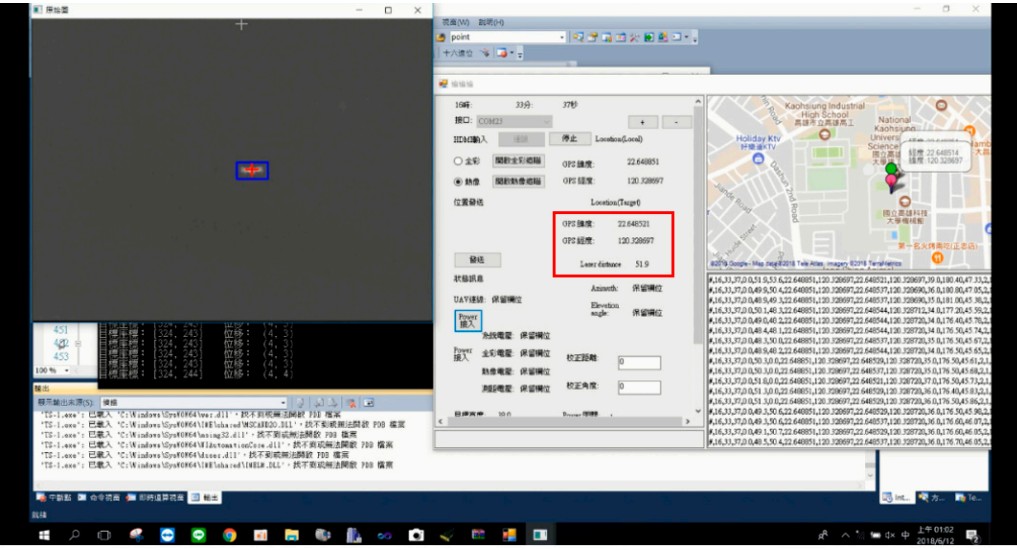

**Figure 18.** Test results of thermal imaging tracing interface.

## 4. Conclusions

As the number of drones increases, so does the risk of flight safety. It is only a matter of time before a drone is used to attack aircrafts to cause damage. Governments are worried that the increasing number of amateur drones will lead to more flight accidents. Therefore, many governments now regulate drone registration, pilot's license regulations, and so forth, and also limit flight range and maximum flight altitude. In addition, various anti-UAV systems have been proposed, but these AUDS either use manual aiming or expensive radar for tracing drones. This study uses two sets of step motors with a thermal imaging camera/full-color camera and sensing module for the dynamic tracing of drone flying at high altitudes, and measurement of its three-dimensional coordinates, which makes it a low-cost and practical drone tracing device for anti-UAV systems. Future research will combine an attack drone and a net gun with a throwing function to provide a safe anti-UAV system with tracing and capturing of drones.

Based on the results of this study in UAV positioning and dynamic locking of the moving coordinates of the UAV, high-energy laser guns or electromagnetic interference guns will be applied for the direct targeting attack in the future work. In addition, real-time transmission of target UAV dynamic coordinate information to an attack UAV will be done for fast pursuit. For the stealth hidden UAV tracking, we will try to search the controller of the UAV remotely by three-point positioning to eliminate the potential threat of the hidden UAV. The future aim of the project is to make the anti-UAV defense system more practical.

**Author Contributions:** Conceptualization, W.-P.C. and B.-H.S.; methodology, W.-P.C. and C.-C.C.; software, C.-I.H.; validation, W.-P.C., B.-H.S. and W.-T.L.; formal analysis, W.-P.C. and B.-H.S.; investigation, C.-I.H.; resources, C.-C.C.; data curation, C.-I.H.; writing—original draft preparation, B.-H.S.; writing—review and editing, W.-P.C.; visualization, W.-T.L.; supervision, C.-I.H.; project administration, W.-P.C.; funding acquisition, B.-H.S. and C.-C.C.

**Funding:** This research was funded by the Ministry of Science and Technology and Southern Taiwan Science Park Bureau of Taiwan, grant number MOST 107-2622-E-992 -006 -CC2 and 107B19.

**Conflicts of Interest:** The authors declare no conflict of interest.

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
