# Peer review of "Development of UAV Tracing and Coordinate Detection Method Using a Dual-Axis Rotary Platform for an Anti-UAV System"

_applsci, doi:10.3390/app9132583_

Round 1

Reviewer 1 Report

The research results are very interesting by proposing a low-cost and practical drone tracing device for anti-UAV systems.

Here are some comments:

-         Enrich the initial part specifying the references and adding literature.

-         Research design and methods details are missing

-         Line 83: it is not clear if you refer to your paper structure, by your system structure

-         Line 83-85: if possible, specify better the scope of the different units composing your system, claryfing the relation existing between them.

-         Check figure 1; some annotations are not readable. In addition, the tracking platform is shown as a detail, but then in the text is not described

-         as a general comment, revise the English form. Some sentences are very long and complicated and not clear.

-         Figures 17 and 18 are not readable

-         Test are described but not critically discussed

Author Response

Response to Reviewer 1 Comments

 The research results are very interesting by proposing a low-cost and practical drone tracing device for anti-UAV systems.

Here are some comments:

-         Enrich the initial part specifying the references and adding literature.

Response : UAVs have only arisen in recent years, and the development of anti-UAV systems has become a national defense level. Therefore, the literature collected in this paper should contain all available documents published by various countries.

-         Research design and methods details are missing

-         Line 83: it is not clear if you refer to your paper structure, by your system structure

-         Line 83-85: if possible, specify better the scope of the different units composing your system, claryfing the relation existing between them.

Response : This research belongs to the product development of national defense application technology, which is a practical application. Therefore, it is really relatively weak in theory, research design and methods.

-         Check figure 1; some annotations are not readable. In addition, the tracking platform is shown as a detail, but then in the text is not described

Response : Figure 1 has been revised.

-         as a general comment, revise the English form. Some sentences are very long and complicated and not clear.

Response : The manuscript was already reviewed by English native speakers.

-         Figures 17 and 18 are not readable

Response : Thanks for the helpful comments.. This paragraph has been revised.

-         Test are described but not critically discussed

Response : Thanks for the helpful comments. This paragraph has been revised.

Reviewer 2 Report

Please think over following recommendations of mine:

1) a 2D mechanism you designed has 2 DoF, better to call them yaw and pitch rotations;

2) Eq (1) introduces a multidimensional function of D(x,y), and it is not clear it is stands for what?! And terms inside of it, stands for what?!

3) Fig 3 has three photos inside, but with the exemption of Fig3(f) the remainings are not expalined. Please eliminate that bottleneck.

4) Line 153: Eq(2) is a little bit messy: it sounds about motor speed, however, it is expressed as pixel difference over time frame of 'Dt'. Please describe your idea more accurate way!

5) Fig 6 depicts a 3D polar coordinate system (yaw angle, pitch angle, distance to target), use that notation freely, it is widely used and accepted.

6) Line 214: you start to discuss solution of air navigation problems in 'large', in spherical coordinate system. Due to your challenge you face to solve, a UAV you are implemented will never fly as HALE UAV, so that chapter, in my opinion, totally useless in the meaning, that you decided to use SAUV for your project. My proposal is totally eliminate that chapter, and, as your future work you can recall for that as extension of your research work to the MALE or HALE UAV categories.

7) Lines 324 and 325: rotational speed is expressed using notation for the speed of the translational motion. Think over, please!

8) Line 334: Please explain, what it is mean: 'four-axis drone': you have ment that 4D in what sense?!

9) How extend your work to those 3D (Dirty-Dull-Dangerous) UAV missions being hidden and illegal?! Discuss it in your conclusions!

Please subject your article for final and very thorough reading to eliminate all weaknesses in grammar, spelling and typing! Lots of work to do here!

Congratulations on your excellent article!

Author Response

Response to Reviewer 2 Comments

Please think over following recommendations of mine:

1) a 2D mechanism you designed has 2 DoF, better to call them yaw and pitch rotations;

2) Eq (1) introduces a multidimensional function of D(x,y), and it is not clear it is stands for what?! And terms inside of it, stands for what?!

3) Fig 3 has three photos inside, but with the exemption of Fig3(f) the remainings are not expalined. Please eliminate that bottleneck.

4) Line 153: Eq(2) is a little bit messy: it sounds about motor speed, however, it is expressed as pixel difference over time frame of 'Dt'. Please describe your idea more accurate way!

5) Fig 6 depicts a 3D polar coordinate system (yaw angle, pitch angle, distance to target), use that notation freely, it is widely used and accepted.

6) Line 214: you start to discuss solution of air navigation problems in 'large', in spherical coordinate system. Due to your challenge you face to solve, a UAV you are implemented will never fly as HALE UAV, so that chapter, in my opinion, totally useless in the meaning, that you decided to use SAUV for your project. My proposal is totally eliminate that chapter, and, as your future work you can recall for that as extension of your research work to the MALE or HALE UAV categories.

7) Lines 324 and 325: rotational speed is expressed using notation for the speed of the translational motion. Think over, please!

8) Line 334: Please explain, what it is mean: 'four-axis drone': you have ment that 4D in what sense?!

9) How extend your work to those 3D (Dirty-Dull-Dangerous) UAV missions being hidden and illegal?! Discuss it in your conclusions!

Please subject your article for final and very thorough reading to eliminate all weaknesses in grammar, spelling and typing! Lots of work to do here!

Congratulations on your excellent article!

Response : Thank you for your valuable comment, the following is a point-by-point response to the questions and comments.

1) Thanks for the helpful comments. The x, y axis has been corrected to yaw and pitch rotation

2) Thanks for the helpful comments. This paragraph has been revised.

3) Thanks for the helpful comments. This paragraph has been revised.

4) Thanks for the helpful comments. This paragraph has been revised.

5) Thanks for the helpful comments.  The relevant terms in this paragraph have been revised.

6)

In this study, the spherical coordinate conversion is mainly to provide the tracking platform system to estimate the UAV's real-time latitude and longitude coordinates, so as to offer the necessary information for attack drone to go to the reference coordinate to capture or destroy the target. After the actual test based on the method of this paper, when the flying height is about 100 meters, the maximum height error is about 2 meters.

This study is limited by funding and regulatory restrictions on the purchase of high-end equipment, such as high-resolution military thermal imaging cameras, high-precision nine-axis attitude meters and long-range precision laser ranges are not easy to obtain.

In the future, if the aforementioned equipment is available, this study will be able to develop a system with the ability of tracking range up to 2000 meters. We will try to use high power laser with automatic adjusting mechanism, and to use automatic zoom camera and thermal imager. The aim is to capture or shoot down UAVs that are at risk of safety at a long distance, to effectively avoid the casualties caused by the UAV.

Thanks for the method provided by the respected reviewer. Follow-up research will be included in our laboratory.

7) Thanks for the helpful comments.  The relevant terms in this paragraph have been revised.

8) Thanks for comments. The four-axis drone refers to the drone of Dajiang.

9) Thanks for comments. This paragraph has been revised.

The manuscript was already revised and reviewed by English native speakers.

This manuscript is a resubmission of an earlier submission. The following is a list of the peer review reports and author responses from that submission.